# The impact of deep learning reconstruction in low dose computed tomography on the evaluation of interstitial lung disease

Chu hyun Kim[1,2], Myung Jin Chung[2,3]*, Yoon Ki Cha[2], Seok Oh[4], Kwang gi Kim[4], Hongseok Yoo[5]

**1** Center for Health Promotion, Samsung Medical Center, Seoul, Republic of Korea, **2** Department of Radiology and AI Research Center, Samsung Medical Center, Sungkyunkwan University, Seoul, Korea, **3** Department of Data Convergence and Future Medicine, Sungkyunkwan University School of Medicine, Seoul, Korea, **4** Gil Medical Center, Department of Biomedical Engineering, Gachon University College of Medicine, Incheon, Korea, **5** Division of Pulmonary and Critical Care Medicine, Samsung Medical Center, School of Medicine, Sungkyunkwan University, Seoul, South Korea

* mj1.chung@samsung.com

**Data Availability Statement:** The datasets generated during and/or analyzed during the current study are available from the corresponding author on reasonable request and if there are legal

## Abstract

To evaluate the effect of the deep learning model reconstruction (DLM) method in terms of image quality and diagnostic agreement in low-dose computed tomography (LDCT) for interstitial lung disease (ILD), 193 patients who underwent LDCT for suspected ILD were retrospectively reviewed. Datasets were reconstructed using filtered back projection (FBP), adaptive statistical iterative reconstruction Veo (ASiR-V), and DLM. For image quality analysis, the signal, noise, signal-to-noise ratio (SNR), blind/referenceless image spatial quality evaluator (BRISQUE), and visual scoring were evaluated. Also, CT patterns of usual interstitial pneumonia (UIP) were classified according to the 2022 idiopathic pulmonary fibrosis (IPF) diagnostic criteria. The differences between CT images subjected to FBP, ASiR-V 30%, and DLM were evaluated. The image noise and BRISQUE scores of DLM images was lower and SNR was higher than that of the ASiR-V and FBP images (ASiR-V vs. DLM, $p < 0.001$ and FBP vs. DLR-M, $p < 0.001$, respectively). The agreement of the diagnostic categorization of IPF between the three reconstruction methods was almost perfect (κ = 0.992, CI 0.990–0.994). Image quality was improved with DLM compared to ASiR-V and FBP.

## Introduction

A precise diagnosis of interstitial lung disease (ILD) is crucial for selecting appropriate treatment candidates [1, 2]. High spatial resolution computed tomography (CT) with thin sections is ideal for evaluating ILD [3, 4]; however, due to concerns about the malignancy risk associated with cumulative exposure to radiation, low-dose chest CT (LDCT) is the preferred modality for conducting follow-up examinations and determining disease progression. However, reducing the radiation dose compromises image quality, which may affect the detection of subtle parenchymal changes [5, 6] and thus, further improvements are needed.

or ethical restrictions on sharing data publicly. Data cannot be shared publicly because of limited anonymity. Data are available from the Department of Radiology and AI Research Center, Samsung Medical Center, Sungkyunkwan University, Seoul, Korea (contact via mj1.chung@samsung.com) for researchers who meet the criteria for access to confidential data.

**Funding:** The author(s) received no specific funding for this work.

**Competing interests:** The authors have declared that no competing interests exist.

To overcome the shortcomings of LDCT, image reconstruction methods after image acquisition can be used to reduce noise, thereby improving image quality and potentially enhancing the diagnostic value [7]. Iterative reconstruction (IR) algorithms are the most widely used CT noise-reduction methods as an alternative to the conventional reconstruction mode based on filtered back projection (FBP) [8–10]. However, IR algorithms are typically nonlinear and task specific, which may modify the spatial resolution and image noise texture [11].

Deep learning technology has been reported to exhibit excellent performance in various fields of medical imaging [10, 12–15]. Deep learning model reconstruction (DLM) has strength in that it does not require simplification of parameters and can handle millions of parameters [16]. A few studies have evaluated image quality and noise in DLM [10, 13–15, 17]; however, to the best of our knowledge, none has specifically investigated its diagnostic impact on ILD.

This study evaluated the effect of the DLM method in terms of quantitative and qualitative image quality and diagnostic impact of DLM in terms of classifying CT pattern in ILD.

## Materials and methods

### Patients

The institutional review board (IRB) of Samsung Medical Center approved this retrospective study (IRB, file number 2021-06-092), and the requirement of informed consent was waived for the use of patient medical data. In total, 369 consecutive patients were included between August 2021 and September 2021. A CT scan was performed because of clinical or radiological suspicion of ILD on chest radiography. All patients underwent routine CT evaluation for ILD, which included helical LDCT and standard-dose non-helical high-resolution CT (HRCT). A total of 170 patients without imaging findings suggestive of ILD (for example, non-dependent ground-glass opacity or reticular abnormalities, non-emphysematous cysts, honeycombing, and traction bronchiectasis) [18] were excluded. Six patients were excluded due to suboptimal image quality. Finally, 193 patients were included in this study.

### CT acquisition and image reconstruction

All CT images were obtained using a multidetector CT scanner (Revolution Frontier, GE Healthcare) under the LDCT protocol without the use of contrast. The protocols consisted of a fixed tube current of 20 mAs per slice (40 mA with a half-second rotation and 0.984 pitch). Slice thickness of 1.25 mm and high-spatial-frequency algorithm were applied. The chest CT protocol used the helical mode with the following parameter: 1.25 mm × 64 detector configuration. The other parameters were as follows: peak tube voltage of 120 kVp, 40-mm table feed per gantry rotation, pitch of 0.984:1, and z-axis tube current modulation. Protocol of LDCT was appropriate according to the guideline of the Korean Society of Thoracic Radiology [19].

Supine inspiratory HRCT scans of all patients were also obtained without intravenous contrast using the same CT scanner for the reference standard. The protocol consisted of sections reconstructed with a high-spatial-frequency algorithm at 1- or 2-cm intervals under automatic exposure control (142–275 mA with dose modulation) with a slice thickness of 1.25 mm, from apex to base.

Three different reconstructions of the LDCT images were obtained: conventional FBP, adaptive statistical iterative reconstruction-Veo at a level of 30% (ASiR-V), and DLM. Images acquired were reconstructed using a FBP algorithm with a sharp convolution kernel and IR algorithms available with vendor's CT scanner for analysis (ASIR; GE Healthcare). For deep learning model reconstruction, FBP images were used and denoised with a dedicated software package (ClariCT.AI, ClariPi). In this study, ASiR-V with blending ratios of 30% was adopted

as a suitable compromise between denoising power and sharpness preservation [9]. And for DLM, since ClariCT.AI provided two tuning parameters; noise blending between 0.0 and 1.0, and edge blending between 0.0 and 1.0, after a pilot visual evaluation, we selected a noise blending value of 0.0 and the edge blending value was set to 0.0. The criteria was two readers' judgement regarding the best balance between image noise and sharpness. All scan data were directly displayed on the picture archiving and communication systems (PACS) (Centricity 3.0, GE Healthcare) workstation monitors. Images were viewed on monitors in lung (width, 1500 HU; level, -700 HU) window settings. To assess radiation exposure in LDCT, we reviewed the CT dose index (CTDIvol) and dose-length product (DLP) recorded as digital imaging and communications in medicine data. The estimated effective dose was calculated as the DLP multiplied by a k-factor of 0.014 $mSv \cdot mGy^{-1} \cdot cm^{-1}$ [20].

## Deep learning reconstruction model

The deep learning model reconstruction (DLM; ClariCT.AI, ClariPi) [13] was developed as a denoising solution using a U-Net-based convolutional neural network. All methods were developed by ClariPi and ClariCT.AI is a commercial product of ClariPi. Contribution of authors was to determine its adequate denoising strength of LDCT for use in ILD assessment. Details are summarized in S1 and S2 Figs in S1 File.

## Image quality analysis

The performance of the image reconstruction methods was evaluated both quantitatively and qualitatively for each case, reconstructed using three different methods (FBP, ASiR-V, and DLM). Quantitative analysis was performed using signal, noise, signal-to-noise ratio (SNR), and blind/referenceless image spatial quality evaluator (BRISQUE) score. For qualitative analysis, a thoracic radiologist visually scored the images.

**Signal, noise, and signal-to noise ratio.** Standardized 20-mm-diameter circular regions of interest were used to record signal and noise, which represented the mean pixel intensity value and standard deviation of pixels for the lung parenchyma, and background air for LDCT scans in FBP, ASiR-V, and DLM image sets [21]. Lung measurements were obtained from the lower lobes towards the periphery to avoid parenchymal lesions. Background air was obtained from the air external and anterior to the patient at the sternomanubrial junction [22]. The signal-to-noise ratio (SNR) was calculated for all three image sets. SNR was calculated as follows [23]:

$$SNR_{background\ air} = |(signal_{background\ air})/(noise_{background\ air})|,\ SNR_{lung\ parenchyma}$$
$$= |(signal_{lung\ parenchyma})/(noise_{lung\ parenchyma})|.$$

**BRISQUE.** BRISQUE is a no-reference image quality assessment model that uses natural scene statistics in the spatial domain [24]. This model is composed of three steps: 1) extraction of natural scene statistics, 2) calculation of feature vectors, and 3) prediction of the image quality score. We utilized a pre-trained prediction model provided by Mittal et al. [24] to predict the image quality score. The minimum and maximum image scores are 0 and 100, respectively, with a lower image score indicating a better image quality. The potential of BRISQUE as an indicator of medical image quality has been reported previously [24–26].

**Visual scoring.** Two thoracic radiologists (reader 1 had 6 years of experience; reader 2 had 15 years of experience) independently assessed CT images and performed a qualitative image analysis on a chest CT scan. The radiologists were blinded to the patients' data and the image reconstruction techniques and examined the images in a random order using PACS.

**Table 1. Visual scoring system used to evaluate image quality.**

| Scale | Contrast | Noise | Overall image quality |
|---|---|---|---|
| 5 | Excellent | Minimal | Best |
| 4 | Above average | Below average | Slight inferior (no influence on diagnosis) |
| 3 | Acceptable | Average | Mildly inferior (possible influence on diagnosis) |
| 2 | Suboptimal | Above average | Moderately inferior (probable influence on diagnosis) |
| 1 | Poor | Unacceptable | Markedly inferior (impairing diagnosis) |

CT scans were graded on axial images with datasets displayed on standard windows, and windowing was allowed, as in routine reporting conditions. The readers randomly assessed the subjective image contrast, noise, and image quality using a five-point visual scoring system, and score of two readers were subsequently averaged [27] (Table 1).

### Evaluating diagnostic agreement on CT pattern diagnosis of ILD

Two thoracic radiologists (reader 1 had 6 years of experience; reader 2 had 15 years of experience) who were blinded to clinical data and image reconstruction techniques independently assessed CT images. Also, images were randomly displayed using PACS to reduce bias. Radiologists determined the radiologic features of usual interstitial pneumonia (UIP), which is a hallmark of idiopathic pulmonary fibrosis (IPF). A classification of 'UIP', 'Probable UIP', 'Indeterminate for UIP', and 'Alternative diagnosis' was assigned for each case using the 2022 American thoracic society and Fleischner Society guidelines [4, 28, 29]. The UIP pattern was defined as subpleural, basal predominance of reticular abnormalities, honeycombing with, or without traction bronchiectasis; the absence of findings was suggestive of alternative diagnosis, including extensive ground-glass opacity, micronodules, discrete cysts, mosaic attenuation, or segmental/lobar consolidation [28]. After diagnosis was made using the three reconstruction methods, cases showing discrepant diagnoses were selected and compared with HRCT findings of the patient. And two radiologists made consensus diagnosis which was considered a reference standard.

### Statistical analysis

Image quality of the three reconstruction methods (FBP, ASiR-V, and DLM) were compared using one-way analysis of variance, and post-hoc pairwise comparisons were adjusted for multiple comparisons using the Bonferroni correction. Average Cohen's kappa statistics were used to evaluate the agreement between the two readers and the diagnostic agreement on the three reconstruction methods. A kappa statistic of 0.81–1.00 indicates an excellent agreement; 0.61–0.80, substantial agreement; 0.41–0.60, moderate agreement; 0.21–0.40, fair agreement; and 0.00–0.20, poor agreement [30]. Statistical significance was set at $p < 0.05$. All statistical calculations were performed using SAS (version 9.4; SAS Institute, Cary, NC, USA) and R (version 3.3.1; R Foundation for Statistical Computing http://www.R-project.org/ [31]) software.

## Results

### Basic characteristics of the participants and radiation dose

Of the 193 patients included in the current study, 141 were men and 52 were women, with a mean age of 68.95 ± 9.39 years (range 36–88 years). A total of 93 patients (48.2%) were diagnosed with IPF based on the diagnostic criteria of the American Thoracic Society and European Respiratory Society [29], 55 patients (28.5%) were diagnosed with connective tissue

disease related ILD, and 19 patients (9.8%) were diagnosed with interstitial lung abnormality. Organizing pneumonia was diagnosed in six patients (3.1%), followed by smoking related ILD in six (3.1%), nonspecific interstitial pneumonia in four (2.1%), and pleuroparenchymal fibroelastosis in four patients (2.1%). Remaining six patients (3.1%) had other diagnoses (for example, chronic hypersensitivity pneumonitis, post inflammatory fibrosis, sarcoidosis, hemosiderosis). Lung biopsy was performed on 38 patients, either wedge resection (34 patients) or transbronchial lung biopsy (4 patients). Of these, 17 were pathologically confirmed to have UIP on surgical lung biopsy.

As for the radiation dosage in LDCT, the mean $CTDI_{vol}$, DLP, and effective dose were 1.96 ± 0.03 mGy, 70.32 ± 5.82 mGy*cm, and 0.98 ± 0.08 mSv, respectively.

## Comparison of imaging reconstruction methods

**Signal, noise, and SNR results.** A comparison of the image signal, noise, and SNR measurements is shown in Fig 1A–1F. The image signal of the lung parenchyma did not significantly differ across FBP, ASiR-V, and DLM; however, the mean background air signal of ASiR-V images was lower than that of FBP and DLM images (FBP vs. ASiR-V, ASiR-V vs. DLM, all $p < 0.001$). The other parameters, including noise and SNR, differed significantly (Fig 2). Image noise was significantly higher in FBP and significantly lower in DLM, both in the lung parenchyma and background air (FBP vs. ASiR-V, ASiR-V vs. DLM, FBP vs. DLM, all $p < 0.001$). The SNR in the lung parenchyma and background air significantly differed across the three different reconstructions, and it was higher in the DLM and lower in the FBP (FBP vs. ASiR-V, ASiR-V vs. DLM, FBP vs. DLM, all $p < 0.001$).

**BRISQUE score results.** The BRISQUE scores of the three reconstructed images are depicted in Fig 1G. The BRISQUE scores of both ASiR-V and DLM were lower than those of the FBP images (all $p < 0.001$). The difference between ASiR-V and DLM was statistically significant ($p < 0.001$), and DLM showed a lower BRISQUE score. The difference between the ASiR-V and FBP images was approximately 1 point, whereas the difference between the DLM and FBP images was approximately 10 points.

**Visual scoring by two thoracic radiologists.** The results of the qualitative image analysis are presented in Table 2. In the evaluation of the image contrast, DLM had highest mean points (4.47 ± 0.52), showing above average to excellent contrast, followed by ASiR-V (3.85 ± 0.56) and FBP (3.62 ± 0.48). The scores significantly differed among the three methods (FBP vs. ASiR-V, ASiR-V vs. DLM, FBP vs. DLM, all $p < 0.001$). In terms of image noise, DLM had highest mean points (3.86 ± 0.45), showing average to below average noise, followed by ASiR-V (3.12 ± 0.78), FBP (2.43 ± 0.78), and scores significantly differed between three methods (FBP vs. ASiR-V, ASiR-V vs. DLM, FBP vs. DLM, all $p < 0.001$). For overall image quality, the DLM images yielded the highest score, which was slightly inferior to the best image quality. The scores between ASiR-V and DLM, ASiR-V and FBP, and DLM and FBP significantly differed (FBP vs. ASiR-V, ASiR-V vs. DLM, and FBP vs. DLM; all $p < 0.001$).

## Diagnostic categorization of UIP based on CT patterns

On the CT pattern diagnosis of UIP, there was substantial agreement between the two readers (κ = 0.617). According to the reference standard that consisted of a consensus panel of two radiologists, the UIP was the most common diagnosis (56.5% in FBP [109 of 193], 56.0% in ASiR-V [108 of 193], and 56.0% in DLM [108 of 193]), followed by probable UIP (19.2% in FBP [37 of 193], 17.6% in ASiR-V [34 of 193], and 18.1% in DLM [35 of 193]), alternative diagnosis (16.6% in all FBP, ASiR-V, and DLM [32 of 193]), and indeterminate for UIP pattern (7.8% in FBP [15 of 193], 9.8% in ASiR-V [34 of 193], and 9.3% in DLM [18 of 193]) (Table 3).

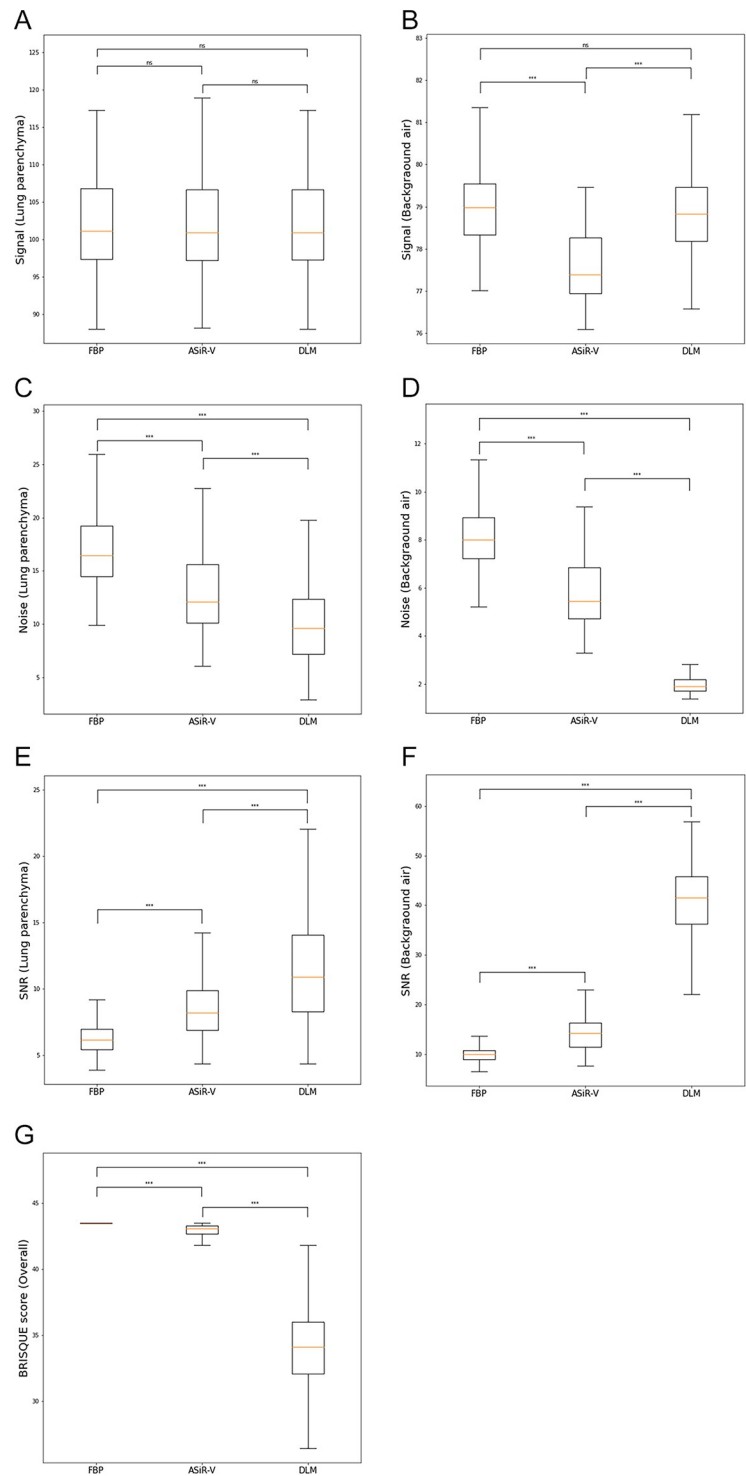

**Fig 1.** Signal (A, B), noise (C, D), signal-to-noise ratio (E, F), and blind/referenceless image spatial quality evaluator score (G) results of three different imaging reconstruction methods (filtered back projection [FBP], adaptive statistical iterative reconstruction-Veo at a level of 30% [ASiR-V], and deep learning image reconstruction [DLM]) at the lung parenchyma and background air. Paired t test results with *** p < 0.01; ns, not significant.

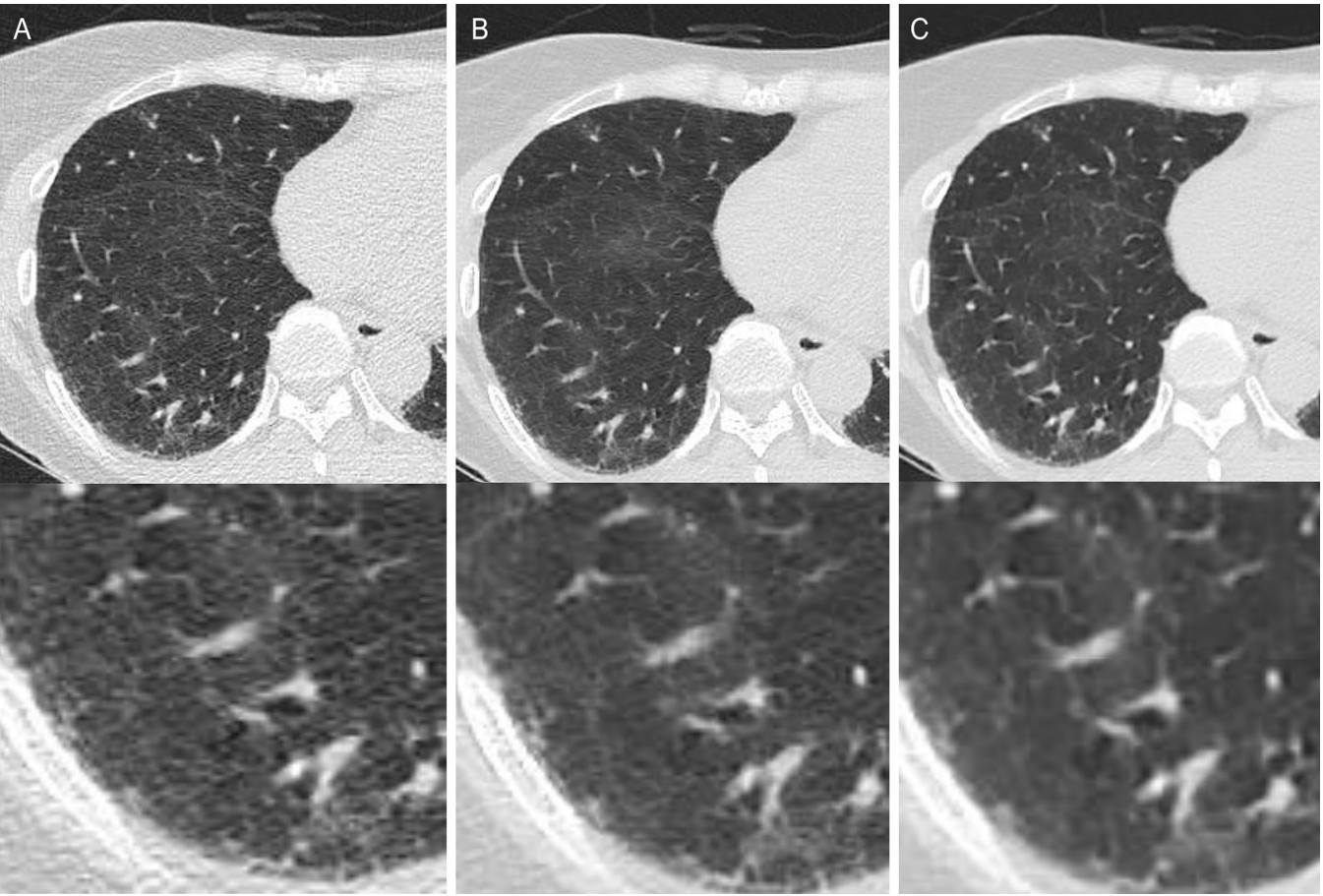

**Fig 2. Comparison of low-dose chest computed tomography scan in axial lung window images of the lung of a 50-year-old man.** Reconstruction performed using filtered back projection (FBP) (A), adaptive statistical iterative reconstruction-Veo at a level of 30% (ASiR-V) (B), and deep learning image reconstruction (DLM) (C). The signal of the lung parenchyma did not vary significantly across the different reconstructions. However, the image noise of the DLM images was lower than that of the ASiR-V and FBP images (ASiR-V vs. DLM, $p < 0.001$ and FBP vs. DLR-M, $p < 0.001$, respectively). The signal-to-noise ratio was significantly higher in DLM in both the lung parenchyma and background air than in other reconstruction methods. (FBP vs. ASiR-V, ASiR-V vs. DLM, and FBP vs. DLM, all $p < 0.001$).

The agreement in diagnostic categorization between the three reconstruction methods was almost perfect (κ = 0.992, CI 0.990–0.994). In the ASiR-V and DLM images, probable UIP was less diagnosed compared to that of FBP (19.2% in FBP [37 of 193], 17.6% in AsiR-V [34 of 193], and 17.6% in DLM [34 of 193]), and indeterminate for UIP was frequently diagnosed (7.8% in FBP [15 of 194], 9.8% in AsiR-V [19 of 193], and 9.8% in DLM [19 of 193]) compared to that of

**Table 2. Comparison of Image quality by visual scoring.**

| Variables | FBP | ASiR-V | DLM | *P* FBP vs. ASiR-V | *P* FBP vs. DLM | *P* ASiR-V vs. DLM |
|---|---|---|---|---|---|---|
| Image contrast | 3.62 ± 0.48 | 3.85 ± 0.56 | 4.47 ± 0.52 | < 0.01 | < 0.01 | < 0.01 |
| Image noise | 2.43 ± 0.78 | 3.12 ± 0.78 | 3.86 ± 0.45 | < 0.01 | < 0.01 | < 0.01 |
| Overall image quality | 3.87 ± 0.98 | 4.01 ± 0.35 | 4.43 ± 0.5 | < 0.01 | < 0.01 | < 0.01 |

FBP = Filtered back projection, ASiR-V = adaptive statistical iterative reconstruction-Veo at a level of 30%, DLM = deep learning image reconstruction.

**Table 3. Diagnostic categorization of usual interstitial pneumonia based on computed tomography patterns.**

| Category | FBP (%) | ASiR-V (%) | DLM (%) |
|---|---|---|---|
| UIP | 109 (56.5) | 108 (56.0) | 108 (56.0) |
| Probable UIP | 37 (19.2) | 34 (17.6) | 34 (17.6) |
| Indeterminate for UIP | 15 (7.8) | 19 (9.8) | 19 (9.8) |
| Alternative diagnosis | 32 (16.6) | 32 (16.6) | 32 (16.6) |

FBP = filtered back projection, ASiR-V = adaptive statistical iterative reconstruction-Veo at a level of 30%,

DLM = deep learning image reconstruction, UIP = usual interstitial pneumonia.

the FBP images (Figs 3 and 4). One case was diagnosed as UIP in the FBP image but categorized as probable UIP in both the ASiR-V and DLM images (Fig 5). There were no discrepant cases in the category of 'alternative diagnosis' between the three reconstruction methods. Cases with discrepant diagnoses among the three reconstruction methods are presented in Table 4.

## Discussion

This study demonstrated that DLM showed favorable results in terms of image noise, SNR, BRISQUE scores, and visual scoring compared to that of the ASiR-V and FBP images. Furthermore, DLM maintained diagnostic agreement in CT pattern diagnosis of UIP.

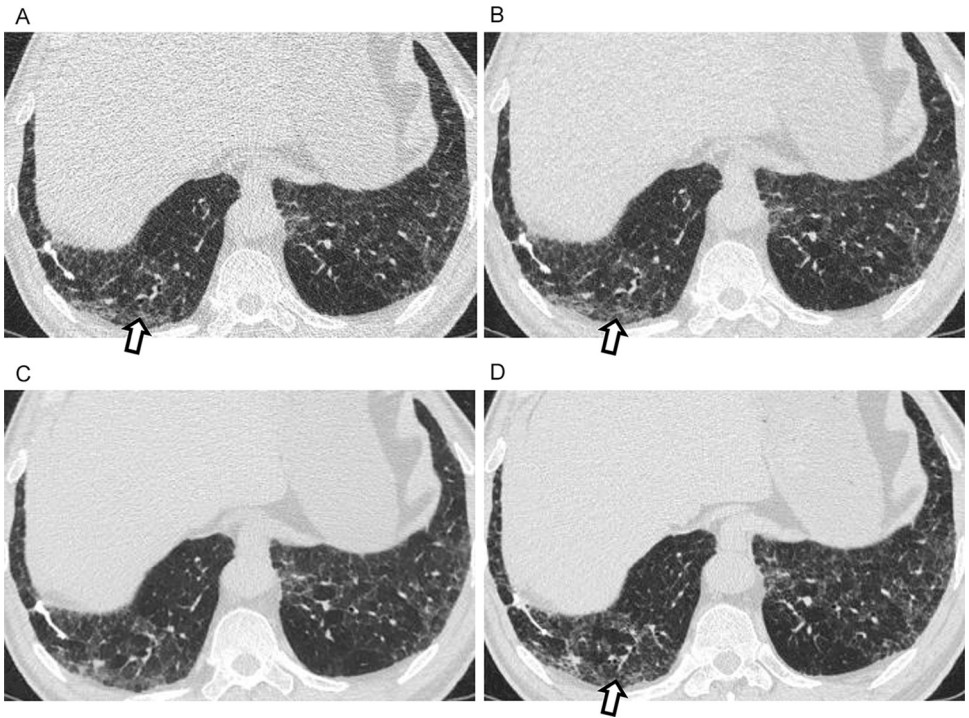

**Fig 3. A discordant case between filtered back projection (FBP), adaptive statistical iterative reconstruction-Veo at a level of 30% (ASiR-V), and deep learning image reconstruction (DLM) in probable usual interstitial pneumonia (UIP) and indeterminate UIP pattern.** CT images of a 66-year-old man showing ground-glass opacity and reticulation in the subpleural areas with a basal predominance (A-D). In the displayed FBP (A) and AsiR-V images (B), readers made the diagnosis of a probable UIP pattern because tubular structures in the periphery of the right lower lobe (arrow) were interpreted as peripheral traction bronchiolectasis. However, in the DLM image (C), the presence of traction bronchiolectasis is not definite. In addition, reticulation is less prominent, and ground-glass opacity is the dominant finding. Therefore, it was categorized as indeterminate for the UIP. In high-resolution CT, (D) probable UIP was suggested because traction bronchiolectasis can be observed (arrow). The patient underwent surgical biopsy, which revealed UIP; therefore, IPF was diagnosed.

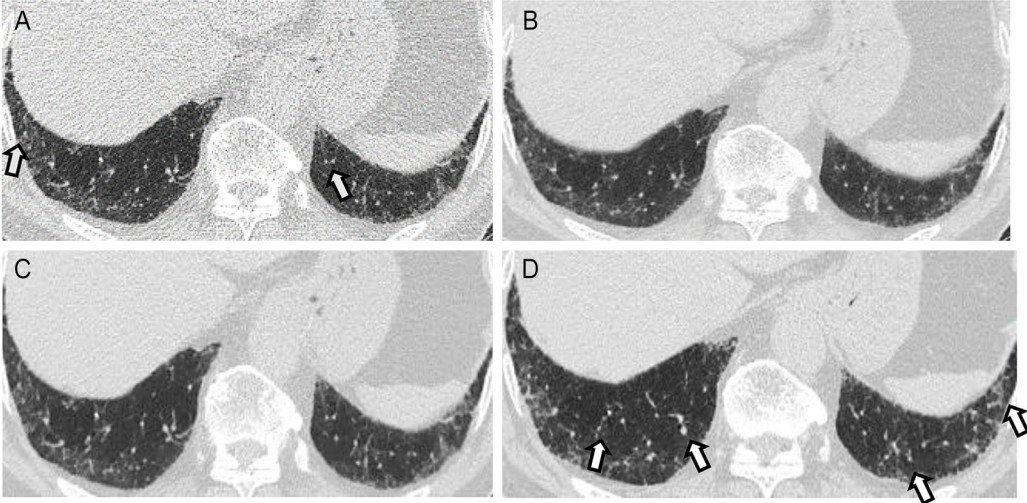

**Fig 4. A discordant case between filtered back projection (FBP), adaptive statistical iterative reconstruction-Veo at a level of 30% (AsiR-V), and deep learning image reconstruction (DLM) in probable usual interstitial pneumonia (UIP) and indeterminate UIP pattern.** CT images of a 61-year-old man showing ground-glass opacity in the basal subpleural areas of the left lower lobe (A-D). In the FBP image (A), reader 1 diagnosed probable UIP pattern, since the observable tubular structures in the periphery (arrows) were interpreted as peripheral traction bronchiolectasis and reticulation was thought to be present. However, reader 2 made the diagnosis of indeterminate for UIP. In the ASiR-V (B) and DLM images (C), two readers diagnosed indeterminate for UIP, since ground-glass opacity with subtle reticulation is dominant without traction bronchiolectasis. In high-resolution CT (D), minimal traction bronchiolectasis can be seen (arrows), and probable UIP was suggested. The final diagnosis of this patient was established using a multidisciplinary approach.

The FBP and IR algorithms are widely used in CT image reconstruction. However, both methods have several drawbacks. FBP can severely degrade image quality by increasing image noise, while IR can modify the spatial resolution and noise texture in different regions of CT images [11, 32]. IR often requires long reconstruction times [11, 27]. With recent advances in artificial intelligence technology, DLM has been introduced to overcome the limitations of FBP and IR approaches. The DLM incorporates convolutional neural networks into the image reconstruction process, which can be used to generate high-quality images from low-dose projection data in a short reconstruction time in a clinical environment [12]. Compared to IR, DLM is a cost effective technology that takes up less space and time, that does not require complex high-level calculations [33].

In this study, we applied the BRISQUE score to assess the image quality of the DLM images with LDCT. Unlike SNR, BRISQUE is a no-reference image quality assessment model based on the principle that natural images possess certain regular statistical properties that are measurably modified by distortions [24]. In our study, DLM images yielded significantly better image quality in the BRISQUE score in concordance with other image parameters, including noise, SNR, and visual scoring.

Recently, several studies have reported that DLM yields a favorable noise texture with superior image quality on low-dose chest CT [15, 17, 22, 23]. However their clinical impact was questionable, because most of studies were performed with normal groups who had clear lung parenchyma without pathology. Less is known whether DLM really works in actual clinical settings.

ILD is a rare condition characterized by extensive inflammation and fibrosis mainly involving the lungs and IPF, the most common type of ILD, is associated with poor prognosis [2, 28]. Since interstitial lung disease (ILD) is characterized by subtle parenchymal changes (e.g.

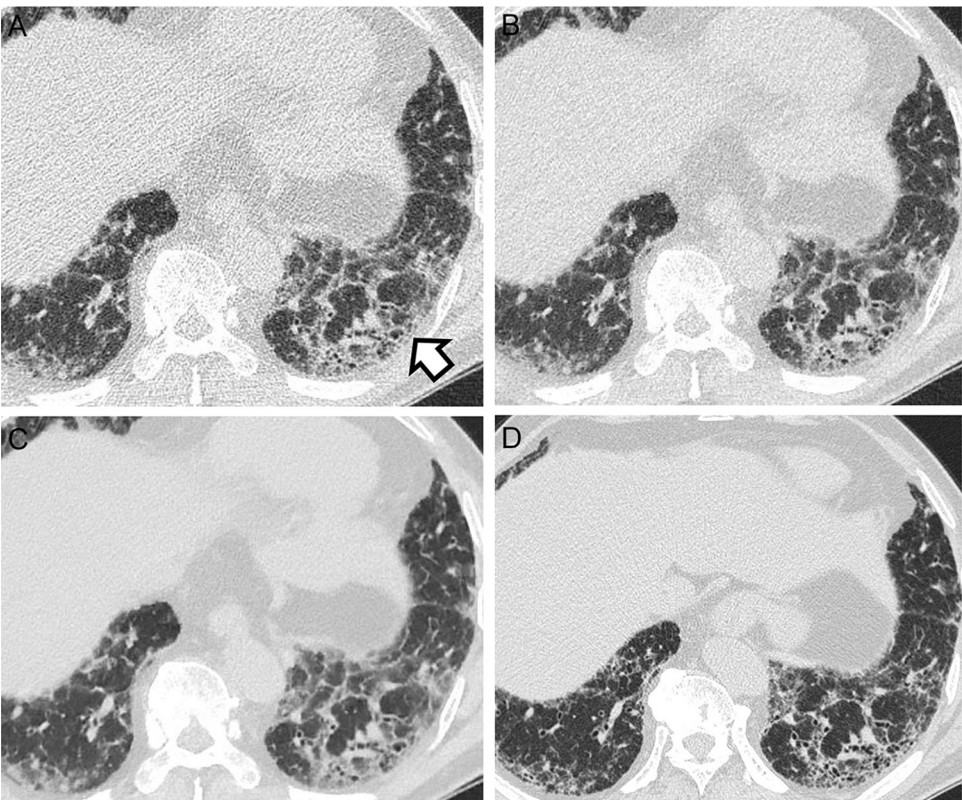

**Fig 5. A discordant case between filtered back projection (FBP), adaptive statistical iterative reconstruction-Veo at a level of 30% (ASiR-V), and deep learning image reconstruction (DLM) in usual interstitial pneumonia (UIP) and probable UIP pattern.** CT images of a 74-year-old man showing subpleural predominant reticular abnormality with traction bronchiectasis (A-D). Clustered, thick-walled cystic spaces are observed in the basal portion of the left lower lobe (arrow), which was interpreted as honeycombing. Therefore, the UIP pattern was diagnosed by two readers using the FBP image (A). However, in the ASiR-V and DLM images, the case was classified as probable UIP by readers because the findings suggesting honeycombing were unclear. On high-resolution CT (D), probable UIP was suggested. The final diagnosis of this patient was established using a multidisciplinary approach.

honeycombing, traction bronchiectasis, ground-glass opacities and reticulation) it may be influenced by image quality largely than other lung etiologies [29]. Diagnosing ILD is challenging for chest radiology specialists and reported inter-/intra- reader variability on ILD is relatively high [4, 9]. Subtle changes in image quality may alter the radiologist's decision. In addition, since ILD is mostly progressive disease, the patients with ILD need continuous

**Table 4. Cases with discrepant diagnosis between three reconstruction methods.**

| Patient Number | FBP | IR | DLM | HRCT diagnosis | Pathology | Final diagnosis |
|---|---|---|---|---|---|---|
| 1 | UIP | Probable UIP | Probable UIP | Probable UIP | None | IPF |
| 2 | Probable UIP | Probable UIP | Indeterminate for UIP | Probable UIP | UIP | IPF |
| 3 | Probable UIP | Indeterminate for UIP | Indeterminate for UIP | Probable UIP | None | Interstitial lung abnormality |
| 4 | Probable UIP | Indeterminate for UIP | Probable UIP | Probable UIP | None | Connective tissue disease related ILD |
| 5 | Probable UIP | Indeterminate for UIP | Indeterminate for UIP | Probable UIP | None | IPF |
| 6 | Probable UIP | Indeterminate for UIP | Indeterminate for UIP | Probable UIP | None | IPF |

FBP = filtered back projection, ASiR-V = adaptive statistical iterative reconstruction-Veo at a level of 30%, DLM = deep learning image reconstruction, UIP = usual interstitial pneumonia, HRCT = high resolution computed tomography, IPF = idiopathic pulmonary fibrosis.

monitoring of the disease and need cancer surveillance by performing routine follow-up CTs. LDCT is widely used to monitor disease progression in ILD patients [5, 6]. Maintaining image quality while using LDCT, thereby minimizing radiation accumulation is paramount for these patients. Therefore, our study aimed to evaluate the image quality by applying DLM to patients with ILD and evaluate whether it maintains an adequate level of diagnostic power compared to other classic imaging reconstruction methods. As a result, in our study, DLM in LDCT maintained diagnostic agreement in terms of CT pattern diagnosis of UIP compared to that of IR and FBP. To the best of our knowledge, this study specifically investigated the diagnostic impact of DLM on ILD, which has never been tried in previous literature.

Although diagnostic agreement between three reconstruction methods was almost perfect, there were six cases (3.1%, 6 of 193) of discrepant diagnoses between the three reconstruction methods (Table 4). Four patients were diagnosed with indeterminate UIP in DLM, while probable UIP was diagnosed on HRCT. Of these, three patients were finally diagnosed with IPF based on multidisciplinary diagnosis. While DLM showed better image quality than that of other reconstruction images, the denoising process may have produced some degradation in the small peripheral airway dilatations or subtle reticulation, which may have caused underestimation of lung fibrosis. Similar results were also observed in IR, with four cases being diagnosed as indeterminate for UIP but probable for UIP on HRCT. Although IR has shown favorable results in the detection and assessment of ILD [8, 9] it is known to have limitations in that the noise texture often differs from that of traditional FBP images and can negatively affect subjective acceptance and diagnostic confidence, altering noise texture [21, 27]. In our study, there were cases in which both IR and DLM were suggestive of such results (Figs 2 and 3). Another possible reason for the discrepant diagnoses is that the reported interobserver agreement in diagnosing UIP pattern is not high ranging 0.40–0.69 [3, 6, 34]. The diagnostic agreement between the two radiologists in this study was substantial; however, variability may have influenced the diagnosis.

It is noteworthy that the CT pattern diagnostic agreement of 'UIP pattern' between the three different reconstructions was almost perfect, except in one case (Fig 4). Diagnosing the UIP pattern by CT is important, since the radiologic diagnosis is sufficient to secure a diagnosis without lung biopsy in an appropriate clinical contest [35, 36]. One discrepant case was found between FBP and other two reconstruction method, and it is thought to have result from substantial noise of the FBP image. Actually, for this reason, the FBP image is seldom used in real clinical practice and considered as raw images when reconstructing CT images. In addition, there were no discrepant cases of alternative diagnoses that may require further diagnostic interventions other than IPF. Most discrepant cases showed differences between 'probable UIP' and 'indeterminate UIP'. CT alone usually is not sufficient to make a definitive diagnosis in these two categories, and additional biopsies or multidisciplinary approaches are needed in most cases. In summary, our study show that diagnostic agreement is maintained in the large framework of DLM in diagnosing UIP pattern by CT, but has not actually shown results that may replace or exceed preexisting methods.

The current study has some limitations. First, our investigation was retrospective. Moreover, the study was conducted at a single institution, which may have caused selection bias. Second, complete blinding of the image reconstruction method was not possible because of the unique visual appearance of the FBP, ASiR-V, and DLM images. Despite the radiologists being blinded to the reconstruction methods and images being randomly displayed in PACS system, it is anticipated that the radiologists were familiar with FBP and ASiR-V images but not with DLM images. Preliminary sessions to let readers get familiar with image appearance before formal analysis could have been helpful to reduce bias. Third, the method we used in our study is original version BRSIQUE which was used for natural images not CT images. A

study using self-developed modified BRISQUE for MRI image evaluation has been published [26]. However, it was difficult to apply in our study because the modified BRISQUE they used were not open to public and our study used different image modality. In our future research, we are planning to develop and apply modified BRISQUE based on pre-trained CT images to increase accuracy.

## Conclusions

In conclusion, the image noise, SNR, BRISQUE, and visual scoring of chest LDCT scan images improved with DLM compared to that with ASiR-V and FBP. DLM may be feasible in clinical practice for evaluating ILD, since diagnostic agreement is maintained in CT pattern diagnosis of UIP compared to that of ASiR-V and FBP.

## Supporting information

**S1 File.**
(DOCX)

## Acknowledgments

Samsung medical center is run by the Samsung life public interest foundation, a social welfare corporation, and is a non-profit medical center.

## Author Contributions

**Conceptualization:** Chu hyun Kim, Myung Jin Chung, Kwang gi Kim, Hongseok Yoo.

**Data curation:** Chu hyun Kim, Seok Oh.

**Formal analysis:** Chu hyun Kim, Seok Oh.

**Project administration:** Chu hyun Kim, Myung Jin Chung, Kwang gi Kim.

**Resources:** Myung Jin Chung, Kwang gi Kim.

**Software:** Myung Jin Chung.

**Supervision:** Yoon Ki Cha, Kwang gi Kim, Hongseok Yoo.

**Validation:** Yoon Ki Cha, Kwang gi Kim, Hongseok Yoo.

**Writing – original draft:** Chu hyun Kim.

**Writing – review & editing:** Chu hyun Kim.

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
