## [Decision Letter · Decision Letter 0]

5 Dec 2022

PONE-D-22-27079The impact of deep learning reconstruction in low dose computed tomography on the evaluation of interstitial lung diseasePLOS ONE

Dear Dr. Chung,

Thank you for submitting your manuscript to PLOS ONE. After careful consideration, we feel that it has merit but does not fully meet PLOS ONE’s publication criteria as it currently stands. Therefore, we invite you to submit a revised version of the manuscript that addresses the points raised during the review process.

We look forward to receiving your revised manuscript.

Kind regards,

Zhentian Wang, Ph.D.

Academic Editor

PLOS ONE

Journal Requirements:

2. Please note that PLOS ONE has specific guidelines on code sharing for submissions in which author-generated code underpins the findings in the manuscript. In these cases, all author-generated code must be made available without restrictions upon publication of the work. Please review our guidelines at https://journals.plos.org/plosone/s/materials-and-software-sharing#loc-sharing-code and ensure that your code is shared in a way that follows best practice and facilitates reproducibility and reuse. New software must comply with the Open Source Definition.

Additional Editor Comments:

Two reviewers have reviewed and commented on your manuscript, and raised several sound concerns. Please address the reviewers' comments point by point if you consider to submit a revision.

Reviewers' comments:

Reviewer's Responses to Questions

**Comments to the Author**

1. Is the manuscript technically sound, and do the data support the conclusions?

Reviewer #1: Yes

Reviewer #2: Partly

2. Has the statistical analysis been performed appropriately and rigorously? 

Reviewer #1: Yes

Reviewer #2: I Don't Know

3. Have the authors made all data underlying the findings in their manuscript fully available?

Reviewer #1: No

Reviewer #2: Yes

4. Is the manuscript presented in an intelligible fashion and written in standard English?

Reviewer #1: Yes

Reviewer #2: Yes

5. Review Comments to the Author

Reviewer #1: This is a well written, well structured and technically sound article addressing an important topic in thoracic CT diagnostics. The pathophysiology of ILD-s, compared to other lung diseases, is still not very well understood and hence their early diagnosis and follow-up remain crucial. The underlying paper investigates their evaluation by means of a DL-based low-dose CT in a retrospective study.

In general, I miss the overall outcome of the paper other than the fact that “DLM” could improve the image quality. I would expect that this is already known from the cited literature and am missing the point of how this is now explicitly relevant to ILDs. Apart from the improved scoring, the data does not seem to show any other benefits in using DLM, compared to the established methods, but maybe that was not the purpose of the stdy. Hence, my first major question to the authors is to iterate more deeply on the relevance of the current study and what exactly was the rationale of it?

Another major point of concern, which I hope the authors can address accordingly is the fact that almost all authors (except one) have affiliation to “Samsung Medical Center”, but at the same time state that they have no competing interest. I am not familiar with Korean law (and thus completely unfamiliar with the above center), but from a science point of view, this cannot be correct:

(a) If “Samsung Medical Center” is a privately owned “for-profit” company/hospital, then the authors should state that in the Competing interests statement. Likewise, if they own and/or receive shares from Samsung and/or any other for-profit institution which could be in relation to the underlying study (e.g. in the form of a gratification/bonus), to my understanding, that should be stated as well.

(b) In case “Samsung Medical Center” is a “non-profit”/”public”/? medical center, then the underlying study to my understanding has been conducted as part of a public/personal funding/grant/fellowship, which should be also accordingly stated under Acknowledgement/Funding and/or Competing Interests as well.

In the “Author Contributions”, I would also have expected that at least all authors have contributed to the “Writing-review & editing” part meaning all authors should have read the manuscript before submission? This was either not the case and/or omitted in the statement (Line 415).

Finally, all other (minor) points are below:

- Line 81 – Line 84: Again, being unfamiliar with Korean Law (and see my major point above), I wonder if the approval of the institutional review board (if that is one of a privately owned company) is enough for ethical clearance and/or does not necessitate further clearance from a University’s (public) ethical board?

- Line 95 – Line 97: Please state whether the LDCT protocol is specific to the used machine and/or published elsewhere, e.g. part of a general guideline?

- Line 106 – Line 114: It is unclear to me whether the three methods were part of the hardware or utilized from some other software packages and/or developed by the authors?

- Line 116 – Line 119: Is it unclear to me which role exactly “ClariCT.AI” plays. Were all methods developed there, what exactly was the authors contribution in those? Does “ClariCT.AI” provide all methods out of the box and/or what development steps (for instance for the U-NET) are really necessary to be conducted by its users? Is the code open-source and/or will it be made available by the authors?

- sFigure 1. has a Typo -- “Convolution” instead of “Convulution”

- Line 126 – Line 129: Can the authors explain why just one Radiologist was used for the image quality (visual scoring) analysis while 2 Radiologists were used for evaluating the diagnostic efficacy?

- Line 190: What is the reference to “Vienna, Austria”? “R” is an open-source programming language and is a community-edition.

- Line 230: The (bottom) images in Figure 2 are either misaligned, deformed and/or some features are missing! That said, yes, the DLM-images exhibit less noise and hence have better contrast, but the images do not exhibit an identical region-of-interest (ROI) and thus cannot be compared.

- This applies to all remaining figures as well. Please make sure that the enlarged regions precisely match so that they can be compared!

- The discussion section misses the points mentioned above (What are the true benefits of DLM? What is the novelty of the present study?)

- Conclusion section is entirely missing in the manuscript

Reviewer #2: This manuscript evaluated the impact of deep learning model reconstruction (DLM) in low-dose computed tomography on the evaluation of interstitial lung disease (ILD), compared to filtered back projection (FBP), and adaptive statistical iterative reconstruction Veo (ASiR-V), using quantitative and qualitative evaluation of imaqe quality and diagnostic efficacy. Based upon a retrospective evaluation with 193 patient cases, authors concluded that DLM improved image quality and maintained diagnostic efficacy in the CT pattern diagnosis of UIP.

While overall well written, a couple of study method and materials need to be clarified.

1. A blind/referenceless image spatial quality evaluator (BRISQUE) was introduced to objectively evaluate image quality. Was it a modified version or original version ? The original BRISQUE was used for natural images, and then a modified version was introduced to adapt to MRI image quality evaluation. The authors need to state the reason for choosing original / modified version and to describe its utility and limitation in the use of CT image evaluation.

2. It is a standard practice to make a tuning of IR and DLM parameters before applying to a specific clinical study. It appears that authors have made tuning of IR and determined to use ASiR-V 30% as a compromise between denoising power and sharpness preservation. However, it is not clear whether the parameter tuning was made for DLM. Which parameters were tuned and what was the criteria ?

3. Manuscript stated that two thoracic radiologists assessed CT images and determined the radiologic features of UIP. It is anticipated that the radiologists were familiar with FBP and IR images but not with DLM images. A standard practice of reader study is to have a preliminary session to let readers get familiar with image appearances. Also important is to avoid reader bias that might arise due to presentation order of certain reconstruction images. Authors did not state any of these. So it is not clear whether the discrepant diagnoses were due to the differences of image familiarity and reader bias. Authors need to clarify these issue.

4. Figures 2 to 5 were presented as examples FBP, ASiR-V, and DLM. In Figures 2, 4 and 5, images were presented in the order of FBP (A), ASiR-V (B), and DLM (C). However in case of Figure 3, the figure legend gives the order a twisted way like FBP (A), DLM (B), and ASiR-V (C). Authors should check if it is a mistake.

5. In Figure 3, which is an example of discordant diagnosis, the image (C) appear to be of a different slice unlike others. Authors should make clear if the readers made the decision based on right images and made records of reconstruction methods correctly.

The lack of clarity and apparent mistakes above invoke a question about the methodological rigorousness of the study. The authors should make careful check of study methods and description in their revision works.

6. PLOS authors have the option to publish the peer review history of their article (what does this mean?). If published, this will include your full peer review and any attached files.

Reviewer #1: No

Reviewer #2: No

---

## [Author Response · Author response to Decision Letter 0]

31 May 2023

Reviewer #1: 

This is a well written, well-structured and technically sound article addressing an important topic in thoracic CT diagnostics. The pathophysiology of ILD-s, compared to other lung diseases, is still not very well understood and hence their early diagnosis and follow-up remain crucial. The underlying paper investigates their evaluation by means of a DL-based low-dose CT in a retrospective study.

1. In general, I miss the overall outcome of the paper other than the fact that “DLM” could improve the image quality. I would expect that this is already known from the cited literature and am missing the point of how this is now explicitly relevant to ILDs. Apart from the improved scoring, the data does not seem to show any other benefits in using DLM, compared to the established methods, but maybe that was not the purpose of the study. Hence, my first major question to the authors is to iterate more deeply on the relevance of the current study and what exactly was the rationale of it? 

Thank you for your suggestion. It seems that our paper lacked sufficient explanation.

Deep learning model reconstruction (DLM) is emerging as a new reconstruction method to replace iterative reconstruction (IR) which incorporates convolutional neural networks into the image reconstruction process. Compared to IR, it is a cost effective technology that takes up less space and time, also cost less and does not require complex high-level calculations (1). As you pointed out, previous studies have studied the effect of DLM on image quality, but its clinical impact was questionable, because most of studies were performed with normal groups who had clear lung parenchyma without pathology. Less is known whether DLM really works in actual clinical settings. We chose interstitial lung disease (ILD) in particular, since the disease is characterized by subtle parenchymal changes (e.g. honeycombing, traction bronchiectasis, ground-glass opacities and reticulation) that may be influenced by image quality largely than other lung etiologies (e.g. cancer, infection) (2). Diagnosing ILD is challenging for chest radiology specialists and reported inter-/intra- reader variability on ILD is relatively high(3, 4). Subtle changes in image quality may alter the radiologist’s decision. In addition, since ILD is mostly progressive disease, the patients with ILD need continuous monitoring of the disease and need cancer surveillance by performing routine follow-up CTs. Maintaining image quality while using low-dose CT thereby minimizing radiation accumulation is paramount for these patients. Therefore, our study is to evaluate the image quality by applying DLM to patients with ILD and evaluate whether it maintains an adequate level of diagnostic power compared to other classic imaging reconstruction methods which has never been tried in previous literature.

Nowadays, numerous AI studies are producing excellent performance in research settings, but clinicians often question its effectiveness when they face their actual performance in real-world practice. In fact, when designing the study we have expected DLM to show results that go beyond IR, FBP in diagnosing ILD. However the results of our study show that diagnostic power is maintained in the large framework of DLM in diagnosing ILD, but has not actually shown results that may replace or exceed preexisting methods. In addition, unexpected disadvantages of DLM was also identified (degrading subtle parenchymal changes while reconstruction). We think out study is sufficiently meaningful and worth publishing, in that it helps understand the application effects and limitations of the current rapidly developing deep learning technology. 

We have incorporated your opinion by adding the above description to the discussion part (Lines 394-418). Thank you

2. Another major point of concern, which I hope the authors can address accordingly is the fact that almost all authors (except one) have affiliation to “Samsung Medical Center”, but at the same time state that they have no competing interest. I am not familiar with Korean law (and thus completely unfamiliar with the above center), but from a science point of view, this cannot be correct:

(a) If “Samsung Medical Center” is a privately owned “for-profit” company/hospital, then the authors should state that in the Competing interests statement. Likewise, if they own and/or receive shares from Samsung and/or any other for-profit institution which could be in relation to the underlying study (e.g. in the form of a gratification/bonus), to my understanding, that should be stated as well.

(b) In case “Samsung Medical Center” is a “non-profit”/”public”/? medical center, then the underlying study to my understanding has been conducted as part of a public/personal funding/grant/fellowship, which should be also accordingly stated under Acknowledgement/Funding and/or Competing Interests as well.

No wonder it feels somewhat unfamiliar to the unique hospital system in Korea. Samsung Medical Center is run by the ‘Samsung life public interest foundation’, a social welfare corporation, and is a non-profit medical center. All authors have no conflict of interest especially with Samsung group. We will clearly address this (Lines 475-478)

3. In the “Author Contributions”, I would also have expected that at least all authors have contributed to the “Writing-review & editing” part meaning all authors should have read the manuscript before submission? This was either not the case and/or omitted in the statement (Line 415).

I'm sorry for the confusion. We have included only major authors in this part. It came from a misunderstanding of authorship policy of journal. Of course, all authors went through the process of reviewing and editing the draft of manuscript. I will add all the authors in this section (included in Cover letter).

4. Line 81 – Line 84: Again, being unfamiliar with Korean Law (and see my major point above), I wonder if the approval of the institutional review board (if that is one of a privately owned company) is enough for ethical clearance and/or does not necessitate further clearance from a University’s (public) ethical board?

According to South Korea’s Bioethics and Safety Act, the establishment of an institutional review board (IRB), as a self-regulatory system, is a requirement for all institutions (including medical centers) performing human research and handling human-derived specimens. There is no need for further clearance from a University’s (public) ethical board in Korea. 

In addition, Samsung medical center institutional review board (IRB) is abided by the pharmaceutical affair law, Bioethics and Safety Act, International Conference on Harmonization (ICH) Good Clinical Practice (GCP) E6(R2) Consensus Guideline and other relevant laws and regulations. 

We have attached our IRB notification letter of the study, which contains more information about Samsung medical center’s IRB system. Thank you. 

5. Line 95 – Line 97: Please state whether the LDCT protocol is specific to the used machine and/or published elsewhere, e.g. part of a general guideline?

Our study protocol of LDCT is appropriate according to the guide line of the Korean Society of Thoracic Radiology (5). We have added the statement in materials and methods section (Lines 77-78). Thank you.

6. Line 106 – Line 114: It is unclear to me whether the three methods were part of the hardware or utilized from some other software packages and/or developed by the authors?

Images acquired were reconstructed using a filtered-back projection (FBP) algorithm with a sharp convolution kernel and Iterative reconstruction (IR) algorithms available with vendor's CT scanner for analysis (ASIR; GE Healthcare). For deep learning model reconstruction, FBP images were used and denoised with a dedicated software package (ClariCT.AI, ClariPi). 

We have reflected your comment by adding more information about three reconstruction methods (Lines 86-90). Thank you.

7. Line 116 – Line 119: Is it unclear to me which role exactly “ClariCT.AI” plays. Were all methods developed there, what exactly was the authors contribution in those? Does “ClariCT.AI” provide all methods out of the box and/or what development steps (for instance for the U-NET) are really necessary to be conducted by its users? Is the code open-source and/or will it be made available by the authors?

Thank you for your kind comments.

Yes, all methods were developed by ClariPi and ClariCT.AI is a commercial product of ClariPi. Contribution of authors was to determine its adequate denoising strength of LDCT for use in ILD assessment. As ClariCT.AI is a commercial product, no further development step was necessary. We have reflected your comment by adding more information in method section. (Lines 108-110).

No, ClariCT.AI is not open-source code.

8. sFigure 1. has a Typo -- “Convolution” instead of “Convulution”

Sorry for our mistake. We modified the typo in sFigure 1. Thank you. 

9. Line 126 – Line 129: Can the authors explain why just one Radiologist was used for the image quality (visual scoring) analysis while 2 Radiologists were used for evaluating the diagnostic efficacy?

Thank you for providing the insights. We added additional radiologist in analyzing visual scoring also. We think the changes are better (Lines 144-153, 259-272)

10. Line 190: What is the reference to “Vienna, Austria”? “R” is an open-source programming language and is a community-edition.

There seems to have been a mistake. You're right. The corresponding part was modified (Lines 187-188).

11. Line 230: The (bottom) images in Figure 2 are either misaligned, deformed and/or some features are missing! That said, yes, the DLM-images exhibit less noise and hence have better contrast, but the images do not exhibit an identical region-of-interest (ROI) and thus cannot be compared.

I'm sorry to have confused you by mistake. Figure 2 has been modified to exhibit identical ROI. 

12. This applies to all remaining figures as well. Please make sure that the enlarged regions precisely match so that they can be compared!

Thank you for careful examination. Adjustment for Figure 4 was made also.

13. The discussion section misses the points mentioned above (What are the true benefits of DLM? What is the novelty of the present study?)

We have emphasized the purpose of the study by adding the answer of question #1 to the discussion part. Thank you (Lines 394-419).

14. Conclusion section is entirely missing in the manuscript

We have separated discussion and conclusion part. Thank you. (Lines 466-472)

Reviewer #2

This manuscript evaluated the impact of deep learning model reconstruction (DLM) in low-dose computed tomography on the evaluation of interstitial lung disease (ILD), compared to filtered back projection (FBP), and adaptive statistical iterative reconstruction Veo (ASiR-V), using quantitative and qualitative evaluation of imaqe quality and diagnostic efficacy. Based upon a retrospective evaluation with 193 patient cases, authors concluded that DLM improved image quality and maintained diagnostic efficacy in the CT pattern diagnosis of UIP.

While overall well written, a couple of study method and materials need to be clarified.

1. A blind/referenceless image spatial quality evaluator (BRISQUE) was introduced to objectively evaluate image quality. Was it a modified version or original version? The original BRISQUE was used for natural images, and then a modified version was introduced to adapt to MRI image quality evaluation. The authors need to state the reason for choosing original / modified version and to describe its utility and limitation in the use of CT image evaluation.

The algorithm used in our study is original BRISQUE not modified BRISQUE. To our knowledge, modified BRISQUE is not a commonly used algorithm and was developed by the authors of the specific study (6). Since the authors did not disclose the pre-trained model with MRI to the public, it was difficult to use unless we develop it ourselves. In my personal opinion, since MRI and CT are distinctly different in modality, it cannot be confirmed that they are unconditionally better.

As for the quality evaluation of CT, we agree with your opinion that it would be better to use a model pre-trained with CT rather than natural image. In our future research, we are planning to develop and apply modified BRISQUE based on pre trained CT images. We will discuss it on limitation section. Thank you for your opinion. (Lines 457-464)

2. It is a standard practice to make a tuning of IR and DLM parameters before applying to a specific clinical study. It appears that authors have made tuning of IR and determined to use ASiR-V 30% as a compromise between denoising power and sharpness preservation. However, it is not clear whether the parameter tuning was made for DLM. Which parameters were tuned and what was the criteria?

Yes, we determined ASiR-V 30% as a suitable compromise between denoising power and sharpness preservation (3). 

ClariCT.AI provided two tuning parameters; noise blending between 0.0 and 1.0, and edge blending between 0.0 and 1.0. After a pilot visual evaluation, we selected a noise blending value of 0.0 and the edge blending value was set to 0.0. The criteria was two readers’ judgement regarding the best balance between image noise and sharpness. We have added these in methods section. (Lines 90-96). Thank you.

3. Manuscript stated that two thoracic radiologists assessed CT images and determined the radiologic features of UIP. It is anticipated that the radiologists were familiar with FBP and IR images but not with DLM images. A standard practice of reader study is to have a preliminary session to let readers get familiar with image appearances. Also important is to avoid reader bias that might arise due to presentation order of certain reconstruction images. Authors did not state any of these. So it is not clear whether the discrepant diagnoses were due to the differences of image familiarity and reader bias. Authors need to clarify these issue. 

The radiologists were blinded to the patients’ data and the image reconstruction techniques and examined the images in a random order using PACS. This statement was included in ‘visual scoring’ section but was omitted in ‘ILD evaluation’ part. Sorry for the mistake. We will add the statement. (Lines 160-161)

However, we fully understand your opinion and the fact that we did not undergone preliminary session is our major limitation of our research. Even though we randomly mixed and analyzed the order of the images, it may not be sufficient. We will clearly state what you have pointed out in the limitation section. (Lines 452-457) Thank you.

4. Figures 2 to 5 were presented as examples FBP, ASiR-V, and DLM. In Figures 2, 4 and 5, images were presented in the order of FBP (A), ASiR-V (B), and DLM (C). However in case of Figure 3, the figure legend gives the order a twisted way like FBP (A), DLM (B), and ASiR-V (C). Authors should check if it is a mistake.

We are sorry for making such a minor mistake. There was a typo between (B) and (C) in figure legend 3 (Lines 319=322). Now it is corrected. Please understand us generously.

5. In Figure 3, which is an example of discordant diagnosis, the image (C) appear to be of a different slice unlike others. Authors should make clear if the readers made the decision based on right images and made records of reconstruction methods correctly.

Thanks for your opinion. However we cannot agree that Figure 3, (C) is a different slice unlike others. We made capture of identical point using spatial cursor function in PACS. The surrounding soft tissue density matches and the calcification in the right lower lobe matches with other slice images. The slight difference in the lung parenchyma seems to have occurred during the image reconstruction process of the DLM. I am deeply sorry for causing this confusion by our mistake in the first place. 

6. The lack of clarity and apparent mistakes above invoke a question about the methodological rigorousness of the study. The authors should make careful check of study methods and description in their revision works.

We are very sorry for our mistakes above. Thank you for giving us the opportunity to strengthen our manuscript with your valuable comments and queries. We have worked hard to incorporate your feedback and hope that these revisions persuade you to accept our submission.

1. Koetzier LR, Mastrodicasa D, Szczykutowicz TP, van der Werf NR, Wang AS, Sandfort V, et al. Deep Learning Image Reconstruction for CT: Technical Principles and Clinical Prospects. Radiology. 2023;306(3):e221257.

2. Lynch DA, Sverzellati N, Travis WD, Brown KK, Colby TV, Galvin JR, et al. Diagnostic criteria for idiopathic pulmonary fibrosis: a Fleischner Society White Paper. Lancet Respir Med. 2018;6(2):138-53.

3. Lim HJ, Chung MJ, Shin KE, Hwang HS, Lee KS. The Impact of Iterative Reconstruction in Low-Dose Computed Tomography on the Evaluation of Diffuse Interstitial Lung Disease. Korean J Radiol. 2016;17(6):950-60.

4. Raghu G, Remy-Jardin M, Richeldi L, Thomson CC, Inoue Y, Johkoh T, et al. Idiopathic Pulmonary Fibrosis (an Update) and Progressive Pulmonary Fibrosis in Adults: An Official ATS/ERS/JRS/ALAT Clinical Practice Guideline. Am J Respir Crit Care Med. 2022;205(9):e18-e47.

5. KSTR. KSTR Chest CT Practice Guidelines and Technical Standards

2007 2008 [Available from: https://kstr.radiology.or.kr/reference/guide.php.

6. Chow LS, Rajagopal H. Modified-BRISQUE as no reference image quality assessment for structural MR images. Magnetic Resonance Imaging. 2017;43:74-87.

---

## [Decision Letter · Decision Letter 1]

14 Jul 2023

PONE-D-22-27079R1The impact of deep learning reconstruction in low dose computed tomography on the evaluation of interstitial lung diseasePLOS ONE

Dear Dr. Chung,

Thank you for submitting your manuscript to PLOS ONE. After careful consideration, we feel that it has merit but does not fully meet PLOS ONE’s publication criteria as it currently stands. Therefore, we invite you to submit a revised version of the manuscript that addresses the points raised during the review process.

Although the quality of the manuscript has been improved after the first-round revision, Reviewer #2 has raised some reasonable concerns which need to be addressed before this work can be published. Please take efforts to address Reviewer #2's comments point by point. 

We look forward to receiving your revised manuscript.

Kind regards,

Zhentian Wang, Ph.D.

Academic Editor

PLOS ONE

Reviewers' comments:

Reviewer's Responses to Questions

**Comments to the Author**

1. If the authors have adequately addressed your comments raised in a previous round of review and you feel that this manuscript is now acceptable for publication, you may indicate that here to bypass the “Comments to the Author” section, enter your conflict of interest statement in the “Confidential to Editor” section, and submit your "Accept" recommendation.

Reviewer #1: All comments have been addressed

Reviewer #2: (No Response)

2. Is the manuscript technically sound, and do the data support the conclusions?

Reviewer #1: Yes

Reviewer #2: No

3. Has the statistical analysis been performed appropriately and rigorously? 

Reviewer #1: Yes

Reviewer #2: No

4. Have the authors made all data underlying the findings in their manuscript fully available?

Reviewer #1: No

Reviewer #2: No

5. Is the manuscript presented in an intelligible fashion and written in standard English?

Reviewer #1: Yes

Reviewer #2: Yes

6. Review Comments to the Author

Reviewer #1: All comments have been adequately addressed. As far as I understand, the data is not published, but can be asked from the authors upon reasonable request.

Reviewer #2: Authors stated that this study aimed to evaluate the effect of DLM method in terms of image quality and diagnostic efficacy of LDCT for ILD, and included 193 suspected ILD patient cases. While authors addressed some of the previous comments, it is hard to capture the study points regarding the definition of efficacy, how it was evaluated, and what reference standard was.

1. Conclusion states "The diagnostic efficacy was maintained in the CT pattern diagnosis of UIP in DLM".

What was the study data supporting this conclusion ? Authors only presented the agreement of the diagnostic categorization of IPF between the three reconstruction methods (FBP, IR, DLM), while none of which is appropriate for reference standard.

2 How was the reference standard established for diagnostic categorization ?

In Lines 303-304, reference standard was made by a consensus panel of two radiologist, which varied depending upon recon methods. And Lines 195-198 state that those readers depended on HRCT findings of the patient to make a reference standard. These are apparently discrepant and cause a confusion.

3. While Lines 331-332 states "There were no discrepant cases in the alternative diagnoses between the three reconstruction methods.", the next sentence (Lines 332-333) says "Cases with discrepant diagnoses among the three reconstruction methods are presented in Table 4." Which one is correct? Again, question arises "What was the main question of the study? Was it evaluating the diagnostic efficacy? Or simply finding agreement between different reconstructions?"

4. Figures 3-5 were used to show the discordant cases between FBP, IR, DLM, and HRCT. How the findings in these Figures relate to study purpose and conclusion ? These Figures seem to convey a contradictory message to the conclusion stating "The diagnostic efficacy was maintained in the CT pattern diagnosis of UIP in DLM".

5. Lines 302-303 states that there was a substantial agreement between the two readers (κ = 0.617). Was it a pooled agreement for the three recons? Was there a better agreement with DLM compared to other recons ?

6. Table 3 compares the diagnostic categorization among the three recon methods without using HRCT, whereas in Table 4 HRCT was used (as a reference ?) along with recon methods. Why were two different ways used ?

Also 'Final diagnosis' appears only in Table 4. Does it mean there was no final diagnosis for the rest 186 cases ? If it was true, then what was reference standard for them and how the diagnostic efficacy was made ?

7. PLOS authors have the option to publish the peer review history of their article (what does this mean?). If published, this will include your full peer review and any attached files.

Reviewer #1: No

Reviewer #2: No

---

## [Author Response · Author response to Decision Letter 1]

3 Aug 2023

Dear. Professor Zhentian Wang

Academic editor of PLOS ONE journal

Thank you for inviting us to re-submit a revised draft of our manuscript. We also appreciate the time and effort you and each of the reviewers have dedicated to providing insightful feedback on ways to strengthen our paper. We have incorporated changes that reflect the detailed suggestions you have graciously provided. 

To facilitate your review of our revisions, the following is a point-by-point response to the questions and comments delivered in the website.

#1. Conclusion states "The diagnostic efficacy was maintained in the CT pattern diagnosis of UIP in DLM".

What was the study data supporting this conclusion? Authors only presented the agreement of the diagnostic categorization of IPF between the three reconstruction methods (FBP, IR, DLM), while none of which is appropriate for reference standard.

Thanks for the good point. As you pointed out, since the reference standard is not clear enough to draw such a conclusion, we have deleted the content and simply state the agreement between different reconstruction methods.

#2 How was the reference standard established for diagnostic categorization ?

In Lines 303-304, reference standard was made by a consensus panel of two radiologist, which varied depending upon recon methods. And Lines 195-198 state that those readers depended on HRCT findings of the patient to make a reference standard. These are apparently discrepant and cause a confusion.

Sorry for the confusion. For discrepant cases, two radiologists referred to HRCT and made a consensus opinion. We will fix that part clearly. (Page 9, Lines 194-200)

3. While Lines 331-332 states "There were no discrepant cases in the alternative diagnoses between the three reconstruction methods.", the next sentence (Lines 332-333) says "Cases with discrepant diagnoses among the three reconstruction methods are presented in Table 4." Which one is correct? Again, question arises "What was the main question of the study? Was it evaluating the diagnostic efficacy? Or simply finding agreement between different reconstructions?"

I am sorry that this part has confused you. The ‘alternative diagnosis’ refers to one of the four classifications of CT pattern diagnosis of UIP. ‘Discrepant diagnosis’ literally refers to cases in which different diagnoses were made between observers. We edited that part to make it more clear. (Page 15, Lines 333-334)

As you pointed out, since the reference standard is not clear enough to draw such a conclusion, we will delete ‘diagnostic efficacy’ part and simply state the agreement between different reconstruction methods.

4. Figures 3-5 were used to show the discordant cases between FBP, IR, DLM, and HRCT. How the findings in these Figures relate to study purpose and conclusion ? These Figures seem to convey a contradictory message to the conclusion stating "The diagnostic efficacy was maintained in the CT pattern diagnosis of UIP in DLM".

Among the four classifications of UIP CT pattern diagnosis, in the case of UIP pattern, all but one discrepant case showed a discrepant opinion. In addition, in the case of 'alternative diagnosis', which requires a completely different diagnosis than IPF, 100% concordant diagnosis was shown.

It is noteworthy that the agreement was almost perfect in these two classifications, since when UIP pattern is radiologically diagnosed in CT, it is sufficient to secure a diagnosis without lung biopsy in an appropriate clinical contest. Discrepant case was found between FBP and other two reconstruction method, and we think that it was result due to too much noise due to the nature of the FBP image (actually, for this reason, the FBP image is seldom used in clinical practice and considered as RAW images when reconstructing CT images). 

In other words, most discrepant cases showed differences between ‘probable UIP’ and ‘indeterminate UIP’. It is difficult to make a definitive diagnosis with CT alone in these two categories, and additional biopsies or multidisciplinary approaches are needed in most cases.

We have embedded this in discussion part. Thank you (Page 21-22, Lines 460-473)

5. Lines 302-303 states that there was a substantial agreement between the two readers (κ = 0.617). Was it a pooled agreement for the three recons? Was there a better agreement with DLM compared to other recons ?

No. We used the average kappa in our analysis. We have addressed it in metod section (Page 10, Line 206). The agreement between each reconstruction was not calculated separately. 

6. Table 3 compares the diagnostic categorization among the three recon methods without using HRCT, whereas in Table 4 HRCT was used (as a reference ?) along with recon methods. Why were two different ways used ?

Table 3 shows the overall diagnosis distribution, and Table 4 was added to describe discrepant cases in more detail.

Also 'Final diagnosis' appears only in Table 4. Does it mean there was no final diagnosis for the rest 186 cases ? If it was true, then what was reference standard for them and how the diagnostic efficacy was made ?

The final diagnosis of the patients are described in the baseline characteristics in result section. (Page 10-11, Lines 221-233)

A total of 93 patients (48.2%) were diagnosed with IPF based on the diagnostic criteria of the American Thoracic Society and European Respiratory Society [29], 55 patients (28.5%) were diagnosed with connective tissue disease related ILD, and 19 patients (9.8%) were diagnosed with interstitial lung abnormality. Organizing pneumonia was diagnosed in six patients (3.1%), followed by smoking related ILD in six (3.1%), nonspecific interstitial pneumonia in four (2.1%), and pleuroparenchymal fibroelastosis in four patients (2.1%). Remaining six patients (3.1%) had other diagnoses (for example, chronic hypersensitivity pneumonitis, post inflammatory fibrosis, sarcoidosis, hemosiderosis). Lung biopsy was performed on 38 patients, either wedge resection (34 patients) or transbronchial lung biopsy (4 patients). Of these, 17 were pathologically confirmed to have UIP on surgical lung biopsy. 

IPF is a clinical diagnosis, and the final diagnosis is made through a multidisciplinary approach in addition to CT finding.

7. PLOS authors have the option to publish the peer review history of their article (what does this mean?). If published, this will include your full peer review and any attached files.

We only follow the journal's publication policy. If you do not want your comments reflected in the publication, we will request the journal to keep them private.

---

## [Decision Letter · Decision Letter 2]

5 Sep 2023

The impact of deep learning reconstruction in low dose computed tomography on the evaluation of interstitial lung disease

PONE-D-22-27079R2

Dear Dr. Chung,

We’re pleased to inform you that your manuscript has been judged scientifically suitable for publication and will be formally accepted for publication once it meets all outstanding technical requirements.

Kind regards,

Zhentian Wang, Ph.D.

Academic Editor

PLOS ONE

Additional Editor Comments (optional):

Reviewers' comments:

Reviewer's Responses to Questions

**Comments to the Author**

1. If the authors have adequately addressed your comments raised in a previous round of review and you feel that this manuscript is now acceptable for publication, you may indicate that here to bypass the “Comments to the Author” section, enter your conflict of interest statement in the “Confidential to Editor” section, and submit your "Accept" recommendation.

Reviewer #2: All comments have been addressed

2. Is the manuscript technically sound, and do the data support the conclusions?

Reviewer #2: Yes

3. Has the statistical analysis been performed appropriately and rigorously? 

Reviewer #2: Yes

4. Have the authors made all data underlying the findings in their manuscript fully available?

Reviewer #2: Yes

5. Is the manuscript presented in an intelligible fashion and written in standard English?

Reviewer #2: Yes

6. Review Comments to the Author

Reviewer #2: All the previous comments were addressed appropriately. The review finds no further comments, and agree the quality of manuscript is now acceptable for publication.

7. PLOS authors have the option to publish the peer review history of their article (what does this mean?). If published, this will include your full peer review and any attached files.

Reviewer #2: No

---

## [Editor Report · Acceptance letter]

18 Sep 2023

PONE-D-22-27079R2 

The impact of deep learning reconstruction in low dose computed tomography on the evaluation of interstitial lung disease 

Dear Dr. Chung:

I'm pleased to inform you that your manuscript has been deemed suitable for publication in PLOS ONE. Congratulations! Your manuscript is now with our production department. 

Kind regards, 

on behalf of

Prof. Zhentian Wang 

Academic Editor

PLOS ONE